# PFFAA: Prototype-based Feature and Frequency Alteration Attack for Semantic Segmentation

## ABSTRACT

Recent research has confirmed the possibility of adversarial attacks on deep models. However, these methods typically assume that the surrogate model has access to the target domain, which is difficult to achieve in practical scenarios. To address this limitation, this paper introduces a novel cross-domain attack method tailored for semantic segmentation, named Prototype-based Feature and Frequency Alteration Attack (PFFAA). This approach empowers a surrogate model to efficiently deceive the black-box victim model without requiring access to the target data. Specifically, through limited queries on the victim model, bidirectional relationships are established between the target classes of the victim model and the source classes of the surrogate model, enabling the extraction of prototypes for these classes. During the attack process, the features of each source class are perturbed to move these features away from their respective prototypes, thereby manipulating the feature space. Moreover, we propose substituting frequency information from images used to train the surrogate model into the frequency domain of the test images to modify texture and structure, thus further enhancing the attack efficacy. Experimental results across multiple datasets and victim models validate that PFFAA achieves state-of-the-art attack performances.

## CCS CONCEPTS

• **Computing methodologies → Image segmentation**.

## KEYWORDS

Black-Box Attack, Cross-Domain Transfer-Based Attack, Semantic Segmentation

## 1 INTRODUCTION

It is well known that computer vision models, such as those employed in classification, object detection, and segmentation, are susceptible to the influence of carefully crafted adversarial examples [4, 16, 31]. In current research, black-box attacks [3, 15, 23, 32] have been extensively studied as a method that limits the information available to attackers. These attacks suppose that the surrogate model is trained on the data related to the victim model, which is challenging in practical scenarios. Some methods [21, 22, 26, 35, 44] focus on the transferability of adversarial samples across domains in classification, e.g., Inkawhich et al. [22] propose a correlation

Permission to make digital or hard copies of all or part of this work for personal or classroom use is granted without fee provided that copies are not made or distributed for profit or commercial advantage and that copies bear this notice and the full citation on the first page. Copyrights for components of this work owned by others than the author(s) must be honored. Abstracting with credit is permitted. To copy otherwise, or republish, to post on servers or to redistribute to lists, requires prior specific permission and/or a fee. Request permissions from permissions@acm.org.

*ACM MM, 2024, Melbourne, Australia*

© 2024 Copyright held by the owner/author(s). Publication rights licensed to ACM.
ACM ISBN 978-x-xxxx-xxxx-x/YY/MM
https://doi.org/10.1145/nnnnnnn.nnnnnnn

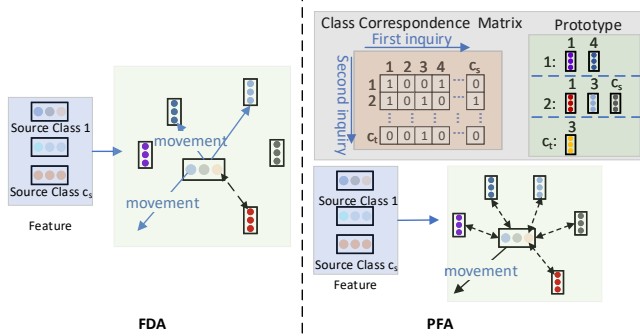

**Figure 1: Difference between FDA and PFA. Compared with FDA, the proposed PFA makes it easier to control the direction of the features in the attack and improve the success rate of the attack.**

matrix-based attack method. However, the cross-domain attacks are still under-explored in semantic segmentation.

Therefore, this paper proposes a cross-domain transferable black-box attack method for segmentation tasks, called Prototype-based Feature and Frequency Alteration Attack (PFFAA). This method addresses the aforementioned issue from two perspectives, i.e., Prototype-based Feature Attack (PFA) and Frequency Alteration Attack (FAA).

Initially, we train a segmentation model (surrogate model) in an independent source domain and establish correspondence between the surrogate and victim models using a limited image set from the source domain. However, the semantic segmentation requires pixel-level predictions, so it is more complex compared with classification. A single target class might correspond to multiple source classes, and these source classes may exhibit significant differences in their feature values. Consequently, the Feature Distribution Attack (FDA) [21] forces the features away from those in clean images, but may steer them towards other intermediate features corresponding to the same target class. To tackle this, we construct a relationship matrix between these classes and devise a feature prototype for each source and target class. Subsequently, we propose modifying the features generated by the surrogate model in the feature space to digress from the corresponding feature prototypes, i.e., PFA. Fig. 1 illustrates the difference between PFA and FDA.

Moreover, given the unknown target domain, we further enhance the effectiveness of the attack through a frequency domain attack, namely the Frequency Alteration Attack (FAA). Specifically, we utilize Fourier transforms to extract detailed information from images in the source domain. Then, we replace the amplitude information of test images in the Fourier space with that from the source domain images and transform them back to the RGB space.

This process alters the texture and structure of the images without any target information.

Extensive experiments on multiple victim models under several different datasets validate the effectiveness of the proposed attack, and the algorithm can successfully fool the victim model without the target data.

Our main contributions can be summarized as follows:

• We present PFFAA, a novel cross-domain attack method that allows for effective attacks on unknown segmentation models without requiring access to the target domain.

• We propose PFA to tackle feature space perturbations that might redirect features towards other features related to the same target class in cross-domain attacks.

• We propose FAA, which modifies the texture and structure of test images by employing Fourier transforms and replacing amplitude information to enhance the effectiveness.

• We demonstrate through extensive experiments that PFFAA generates transferable adversarial samples with significantly better performances than state-of-the-art methods.

## 2 RELATED WORK

### 2.1 Black-box Adversarial Attacks

In black-box attacks, the attacker has no access to the model and cannot obtain model parameters or gradients computed by back-propagation. Currently, black-box attack methods fall into two categories: transfer-based attacks [7, 24, 28, 36] and query-based attacks [6, 10, 20, 25, 39]. The former assumes similarity between the model and the victim model, allowing adversarial examples generated by a surrogate model to deceive the victim model. The latter explores the adversarial space and generated attacks based on feedback obtained from the victim model. While these methods often achieve high success rates, they might require a substantial number of queries.

### 2.2 Cross-Domain Attacks

Several recent studies have proposed methods for cross-domain attack capabilities for transfer-based attacks. There are two approaches to generate perturbations: decision space and feature space attacks. The former aims to influence the output layer of classifiers, pushing predictions away from the correct decision boundary. This is typically achieved by optimizing cross-entropy loss. Dong et al. [11] introduced a momentum-based iterative algorithm to strengthen the attack effect. Additionally, they [12] proposed a translation-invariant method to enhance the transferability of adversarial examples. Xie et al. [43] randomly transformed input images at each iteration to increase diversity. The latter generates perturbations primarily by moving features away from the original state. Ganeshan et al. [14] identified the limitations of decision space attacks and introduced a new attack called FDA. Inkawhich et al. [21] developed a novel attack based on modeling classification and layer depth feature distribution. While Lu et al. [30] and Naseer et al. [34] destroyed intermediate features of models to modify images, they focused on attacking distinct visual tasks rather than cross-domain settings. Inkawhich et al. [22] proposed a correlation matrix-based attack. Wang et al. [41] trained the surrogate model from scratch by the adversary-centric contrastive learning with unlabeled data. However, these methods are designed for classification tasks, and cross-domain and cross-model attacks on semantic segmentation have not been fully investigated.

### 2.3 Attacks against segmentation

Different from classification tasks, semantic segmentation possesses higher task complexity [1, 2, 19, 42]. Cai et al. [4] employed a collection of context-aware attacks based on proxy sets and queries. Recent research [1, 15, 16] introduced a transferable non-targeted attack using a single proxy model. SegPGD [16] can effectively utilize adversarial strategies to mislead the prediction results. CosPGD [1] incorporated attacks for pixel prediction tasks and exploited the cosine similarity between the prediction and the ground truth. In contrast to these individual proxy models, EBAD [3] used multiple agent models to generate more effective adversarial examples. Rony et al. [37] handled large numbers of constraints within a non-convex minimization framework via an Augmented Lagrangian approach, coupled with adaptive constraint scaling and masking strategies. However, these methods do not consider cross-domain transferability of adversarial samples.

### 2.4 Frequency-based analysis

Recent studies have analyzed the adversarial example approach through a frequency domain perspective and found that the low-frequency components primarily represent image content. In contrast, high-frequency components encode edge and texture information. Wang et al. [40] utilized high-frequency features to improve the accuracy of models, suggesting a smoothing convolution kernel approach. Conversely, Guo et al. [17] proposed the attack method, which targets low-frequency components to reduce model queries. However, the generated adversarial examples are not realistic enough and are easily detected. Recent works have concentrated on manipulating specific frequency content to craft adversarial attacks. Deng et al. [9] proposed the perturbation generation across frequency domains. Maiya et al. [33] introduced the frequency-based analysis for balancing accuracy and robustness. However, these approaches do not address the complexity of semantic segmentation attacks.

## 3 METHODOLOGY

### 3.1 Overview

Let $x_t$ be a colored image with a size $W \times H$, and $y_t$ is the ground-truth label in the target domain $D_t$. The victim model $M_t$ is trained on $D_t$, which includes a feature extractor $h_t$ and a classifier $f_t$. Our goal is to train a surrogate model $M_s$ on the source domain $D_s$ that derives the adversarial example $x'_t = x_t + \delta$, where $\delta$ is the perturbation generated by the attack algorithm. This adversarial example will induce the victim model to make incorrect predictions. To make sure that the difference between $x'_t$ and $x_t$ is imperceptible, the $l_\infty$ norm of the perturbation $\delta$ is constrained to be smaller than a threshold $\epsilon$, i.e., $||x'_t - x_t||_\infty < \epsilon$. Therefore, the final optimization problem can be formulated as:

$$M_t(x'_t) \neq y_t,$$
$$s.t. \ x'_t = \arg\max_{x'_t} \mathcal{L}(M_s, x_t), ||x'_t - x_t||_\infty < \epsilon \quad (1)$$

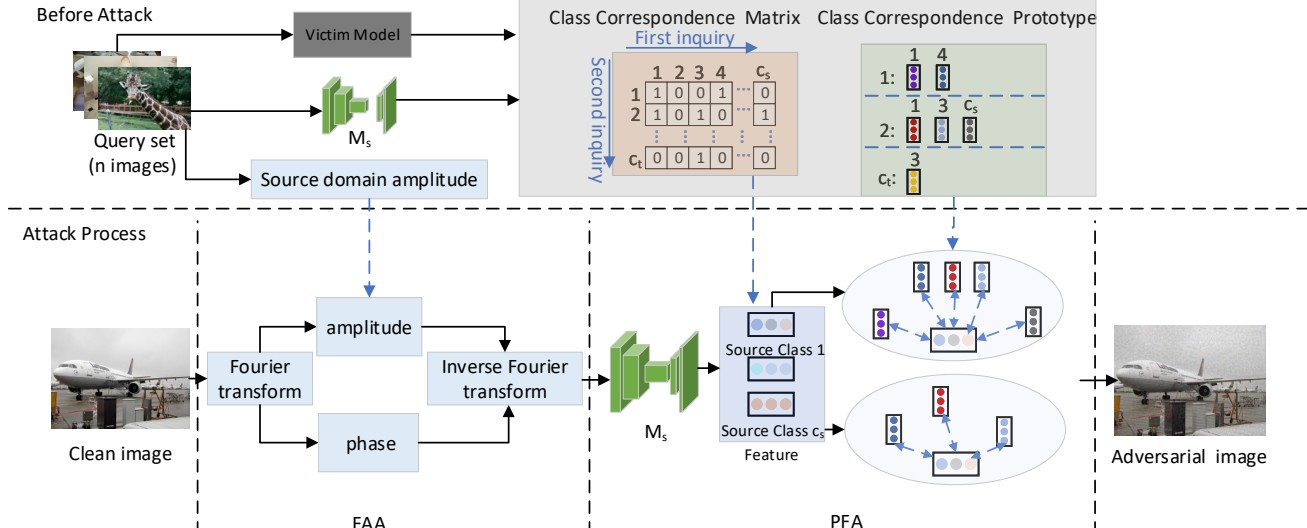

Figure 2: Overview of PFFAA. Our method obtains the correlation matrix, the prototype, and the Fourier amplitude information of the source data in the query amplitude. After that, it generates the adversarial image by realizing the perturbation of the image through FAA and PFA in the attack phase.

where $\mathcal{L}(M_s, x_t)$ means the objective function of the attack strategy, which is used to generate the adversarial image $x_t'$ by $M_s$. $\epsilon$ means the perturbation budget.

We employ PGD to optimize the perturbation, which is an untargeted attack:

$$\delta^{t+1} = \delta^t + \lambda \nabla_\delta sign(\mathcal{L}(M_s, x_t)) \qquad (2)$$

where $t$ indicates the attack step, and $\lambda$ is the step size.

Fig. 2 shows the proposed method. Initially, before the attack on a victim model, we conduct several queries to obtain the Class Correspondence Matrix and Class Correspondence Prototype, which is used to establish the relationships between classes. Furthermore, based on the source domain image, we extract the amplitude features of the source domain in Fourier space. It is worth noting that this step is a one-time process and can be considered as an offline operation. Subsequently, our method conducts a two-stage attack on clean images. In the first stage, we use the amplitude features of the source domain in Fourier space to replace the amplitude features of the clean image and obtain a new image. In the second stage, the features of the new image are pushed away from the prototypes of relevant classes in the feature space. The proposed method will be further elaborated in the following subsections.

## 3.2 Extract Correlation Information

Before the formal attack, we establish a relationship between the victim and surrogate models by conducting a limited number of queries to the victim model. Specifically, we select a query set $\mathbb{S}_q$ from the source domain $D_s$, containing $n$ images. We then obtain the prediction map for each image, indicated as the pseudo-label $y_t^*$ on the target domain by $M_t$. Additionally, these images are queried by $M_s$ to obtain the pseudo-label $y_s^*$ on the source domain. With these sets of pseudo-labels, we can obtain the relationship matrix

$I \in \mathbf{R}^{\mathbb{C}_s \times \mathbb{C}_t}$, where $\mathbb{C}_t$ represents the set of classes on the source domain, and $\mathbb{C}_s$ represents the set of classes appearing in the query on the target domain. For a pair of classes $I_{c_s, c_t}$, the value $I_{c_s, c_t}$ equals 1 in the relationship matrix if the surrogate model predicts the presence of pixels as $c_s$ and the victim model predicts it as $c_t$. This creation of relationships is accomplished by querying over different domains and models based on a small number of queries. Additionally, we extract the feature prototype $\mathbf{p}_{\mathbf{c}_s, \mathbf{c}_t}$ on the surrogate model for each $(c_s, c_t)$:

$$\mathbf{p}_{\mathbf{c}_s, \mathbf{c}_t} = \frac{\sum_{s_q \in \mathbb{S}_q} \sum_i^{HW} h_s(s_q) \mathbb{1}_{y_t^* = c_t \cap y_s^* = c_s}}{\sum_{s_q \in \mathbb{S}_q} \sum_i^{HW} \mathbb{1}_{y_t^* = c_t \cap y_s^* = c_s}} \qquad (3)$$

where $h_s$ is the feature extractor.

The $\mathbf{p}_{\mathbf{c}_s, \mathbf{c}_t}$ represents the features of the corresponding surrogate model when the two models predict $c_s$ and $c_t$, respectively.

Furthermore, to implement FFA, we start the process by extracting and fusing amplitude information from the $n$ images of the source domain. The purpose of this fusion is to create a combined representation of the frequency components from multiple source images, denoted as:

$$F_S(u, v) = \frac{1}{|\mathbb{S}_q|} \sum_{s_q \in \mathbb{S}_q} \mathcal{F}(s_q[m, n]) \qquad (4)$$

where $F_S(u, v)$ denotes the fused frequency information from the source domain, $u$ and $v$ represent the frequency components, and $\mathcal{F}$ is Fast Fourier transform. $m$ and $n$ represent the row and column indices of a image, respectively.

## 3.3 Prototype-based Feature Attack

Feature space methods use white-box intermediate feature information to compute the adversarial perturbation. In contrast, PFFAA leverages the intermediate feature information of the surrogate

---

**Algorithm 1** Attack process of each image

**Input**: Surrogate model $M_s$, and clean image $x_t$
**Output**: Attack image $x'_t$
1: Get pseudo-label $y^*_t$ and feature $F_t$ of $x_t$ from $M_s$
2: Get the image $x_{s->t}$ with Eqs. (6), (7) and (8)
3: Clip $x_{s->t}$ with $||x_{s->t} - x_t||_\infty < \epsilon$
4: **while** step less than total attack steps **do**
5:    Get feature $F'_{s->t}$ of $x_{s->t}$ from $M_s$
6:    Compute loss $\mathcal{L}_{PFA}$ according Eq. (5)
7:    Update $\delta$ with Eq. (2)
8:    Get $x'_t$ by $x_{s->t} + \delta$
9:    Clip $x'_t$ with $||x'_t - x_t||_\infty < \epsilon$, and $x_{s->t} = x'_t$
10: **end while**
11: Return $x'_t$

---

model $M_s$ to compute this adversarial perturbation. The multi-layer Feature Distribution Attack (FDA) is often used in feature space attacks. However, FDA does not guarantee the effectiveness of cross-domain attacks. Therefore, we design a multi-layer Prototype-based Feature Attack (PFA). Specifically, for each source class, PFA obtains the set of possibly related source classes by two queries of the matrix. PFA first acquires the possible corresponding target classes of the source classes, and then gets the possible associated source classes of each target class to form a set $(S_{c_s,c_t})$. The loss function of PFA $\mathcal{L}_{PFA}$ is defined as:

$$\mathcal{L}_{PFA} = \sum_c^{y^*_t}(p(c|h^c_s(x+\delta)) + \eta\,(\frac{||h^c_s(x+\delta) - h^c_s(x)||_2}{||h^c_s(x)||_2} \\ + \sum_{(c_s,c_t)\in(S_{c_s,c_t})} \frac{||h^c_s(x+\delta) - \mathbf{p}_{\mathbf{c_s},\mathbf{c_t}}||_2}{||\mathbf{p}_{\mathbf{c_s},\mathbf{c_t}}||_2})) \tag{5}$$

where $h^c_s(\cdot)$ means the features associated with class $c$, and $p(\cdot)$ denotes the predicted probability. $\eta$ is the weight.

## 3.4 Frequency Alteration Attack

Frequency domain attack is a technique that leverages the spectral information of an image, obtained through a two-dimensional Fourier transform, to manipulate the structure and features of the image with targeted low-frequency modifications. The process begins by transforming the target image into the frequency domain to extract its amplitude and phase information. We convert the clean image $x_t$ to the frequency domain by Fast Fourier transform:

$$x_t(u,v) = \mathcal{F}(x_t[m,n]) = \sum_{m=0}^{H-1}\sum_{n=0}^{W-1} x_q[m,n]\cdot e^{-i2\pi(\frac{um}{H}+\frac{vn}{W})} \tag{6}$$

where $x_t(u,v)$ is the Fourier transform of the clean image.

Therefore, the amplitude of the query image is $\phi_c(u,v) = |x_t(u,v)|$. Subsequently, the amplitude information extracted from the source domain is applied to replace the amplitude of the target image. This selective replacement is realized by the following equation:

$$\phi_c(u,v) = \phi_S(u,v) \tag{7}$$

where $\phi_S(u,v)$ is the source amplitude, which equals $|F_S(u,v)|$.

Finally, the modified amplitude is transformed back to the original image space using the inverse Fourier transform to obtain the

image $x_{s->t}$:

$$x_{s->t} = Clip_\epsilon(\mathcal{F}^{-1}(\phi_c(u,v), arg(x_t(u,v)))) \tag{8}$$

where $\mathcal{F}^{-1}$ is the inverse Fourier transform, $Clip_\epsilon$ means the clipping operation, and $arg(*)$ means the phase of it.

This process changes the amplitude information of the target image, thus changing the texture and structure of the image. It is worth noting that the image $x_{s->t}$ still needs to conform to the constraint $||x'_t - x_t||_\infty < \epsilon$.

## 3.5 Attack Process

Algorithm 1 shows the pseudo-code of the attack process for each image. PFFAA first attacks the clean image using the frequency information of the source domain to change its texture and other information. After that, the features and predictions output by the image on the surrogate model are obtained and the image is attacked using the proposed prototype-based feature attack method to obtain adversarial samples that are transferable over different domains and model structures. Note that the optimization loss $\mathcal{L}(M_s, x_t)$ in Eq. (1) is $\mathcal{L}_{PFA}$ in our attacker.

## 4 EXPERIMENTS

### 4.1 Experimental Setup

**Datasets and victim models.** To verify the effectiveness of the proposed method, we select multiple target domains and various models to validate its ability to generate adversarial samples with strong transferability. Specifically, we conduct experiments on validation datasets obtained from Pascal VOC2012 [13], ADE20k [46] and Cityscapes [8], which contain 1499, 2000, and 500 images with 21, 150, and 19 classes, respectively. For the black-box victim models, we choose six different models, including PSPNet [45] and DeepLabV3 [5] with ResNet50 [18] as the backbone (referred to as PSP-R50 and DLV3-R50, respectively), PSPNet and DeepLabV3 with ResNet101 [18] as the backbone (referred to as PSP-R101 and DLV3-R101, respectively), UperNet with ResNet101 as the backbone, and HRNet [38] with FCN [29] as the decoder and HRNet48 [38] as the backbone.

**Implementation details.** For surrogate models, we choose PSP-Net and DeepLabV3 with ResNet50 as the backbone. The source domain $D_s$ is the COCO 2017 dataset [27] in our experiments, which contains 164k images and 171 semantic categories. The surrogate model is trained on the COCO 2017 dataset with 80k steps. We use the perturbation budget $l_\infty \le 16$ out of 255 in PGD. As in previous work [22, 26], the number of iterations of the PGD attack is 10. We set $n = 100$ in our experiments. The hyperparameter $\eta$ is set to 10.

**Competitors.** We adopt the recent cross-domain transfer-based attacks as baselines. Specifically, we utilize attackers such as BIA [44], CDA [35], CDTA [26], AGS [41], and the attacker proposed by Inkawhich et al. [22]. For BIA, CDA, CDTA and AGS, we use their proposed models as surrogate models. While Inkawhich et al. [22] propose the Class Correlation Matrix attack for cross-domain scenarios, which is model-agnostic, so we retrain DLV3-R50 as the surrogate model for this attacker. Additionally, we incorporate PGD [31] and FDA [14] to show the challenges of cross-domain and cross-model attacks. They generate adversarial images from surrogate models (DLV3-R50 and PSP-R50) trained on the COCO dataset. For

**Table 1: Comparisons of attack mIoU scores (%) on the Pascal VOC2012 dataset. "Surrogate model" is the surrogate model used by each attacker, which is a white-box network for the attacker. Herein, the best results are marked in boldface.**

| Attack | Surrogate model | Black-box Victim model (mIoU ↓) | | | | | |
|--------|-----------------|---------|--------|----------|----------|--------|-------|
| | | DLV3-R50 | PSP-R50 | DLV3-R101 | PSP-R101 | UperNet | HRNet |
| Clean Images | - | 76.17 | 76.78 | 78.70 | 78.47 | 77.10 | 75.87 |
| BIA [44] | - | 50.08 | 50.32 | 51.29 | 52.56 | 56.90 | 56.43 |
| CDA [35] | - | 47.71 | 46.60 | 51.71 | 51.72 | 52.06 | 53.13 |
| CDTA [26] | - | 45.32 | 44.70 | 49.89 | 50.46 | 51.81 | 51.35 |
| AGS [41] | - | 54.28 | 53.50 | 56.90 | 57.11 | 55.98 | 57.83 |
| Inkawhich et al. [22] | DLV3-R50 | 32.38 | 31.64 | 33.35 | 32.64 | 40.29 | 42.55 |
| PGD [31] | DLV3-R50 | 38.59 | 38.83 | 42.74 | 43.55 | 42.33 | 50.79 |
| | PSP-R50 | 39.39 | 38.97 | 42.40 | 43.27 | 44.09 | 51.31 |
| FDA [14] | DLV3-R50 | 35.66 | 36.99 | 38.42 | 39.02 | 42.71 | 47.15 |
| | PSP-R50 | 34.30 | 37.52 | 38.10 | 38.24 | 43.29 | 49.07 |
| PFFAA (Ours) | DLV3-R50 | 22.96 | 23.93 | **23.31** | 24.87 | 20.61 | 25.91 |
| | PSP-R50 | **22.33** | **21.58** | 24.14 | **22.91** | **20.51** | 25.35 |

**Table 2: Comparisons of attack mIoU scores (%) on the ADE20k dataset.**

| Attack | Surrogate model | Black-box Victim model (mIoU ↓) | | | | | |
|--------|-----------------|---------|--------|----------|----------|--------|-------|
| | | DLV3-R50 | PSP-R50 | DLV3-R101 | PSP-R101 | UperNet | HRNet |
| Clean Images | - | 42.42 | 41.13 | 44.08 | 41.90 | 43.57 | 41.90 |
| BIA [44] | - | 33.71 | 34.48 | 36.84 | 37.05 | 35.86 | 38.36 |
| CDA [35] | - | 28.91 | 29.18 | 31.07 | 31.78 | 29.78 | 35.38 |
| CDTA [26] | - | 26.44 | 27.74 | 28.62 | 29.63 | 27.75 | 32.18 |
| AGS [41] | - | 22.25 | 21.48 | 23.07 | 21.48 | 21.70 | 23.69 |
| Inkawhich et al. [22] | DLV3-R50 | 18.30 | 16.80 | 20.16 | 20.97 | 19.94 | 25.60 |
| PGD [31] | DLV3-R50 | 22.67 | 21.23 | 25.43 | 24.58 | 23.67 | 27.17 |
| | PSP-R50 | 22.55 | 21.35 | 24.97 | 24.11 | 23.13 | 27.44 |
| FDA [14] | DLV3-R50 | 20.74 | 20.48 | 23.46 | 22.78 | 23.14 | 25.15 |
| | PSP-R50 | 21.00 | 19.65 | 23.13 | 22.21 | 23.01 | 24.28 |
| PFFAA (Ours) | DLV3-R50 | **7.62** | **9.60** | **11.01** | **12.09** | **9.85** | 14.78 |
| | PSP-R50 | 8.54 | 10.97 | 12.56 | 12.62 | 11.34 | **13.55** |

competitors [21, 22, 26, 31, 35, 44], we give further instructions to explain our experimental setup. The surrogate models of BIA, CDA, CDTA, and AGS are obtained by training them after careful design, mainly to make the models generate universal intermediate features. Therefore, for these four methods, we use their original models as surrogate models.

**Evaluation metrics.** For semantic segmentation, the mean Intersection over Union (mIoU) metric is frequently used to measure the performance of the model. Therefore, the attack performance is evaluated using mIoU, and the lower the mIoU score the better the attack performance.

## 4.2 Quantitative Evaluation

We first investigate the effectiveness of the proposed method compared with competitors.

**Pascal VOC2012 dataset.** Table 1 presents the experimental results on the Pascal VOC2012 dataset. Notably, all victim models demonstrate effective performance on clean images. Initially, algorithms designed for cross-domain classification attacks (BIA, CDA and CDTA) frequently disregard the complexities of semantic

segmentation, leading to less effective attack results. AGS performs below the desired level because its surrogate model is trained on the unlabeled dataset. FDA that attacks in the feature space, clearly outperforms PGD, which attacks in the decision space, achieving approximately 2% higher attack performance. This highlights the importance of prioritizing the feature space in these types of attacks. Moreover, Inkawhich et al. [22] successfully outperforms FDA using DLV3-R50 as a surrogate model. It is worth noting that these methods are substantially less effective in attacking HRNet. This is because HRNet employs HRNet48 as the backbone, which generates significantly different intermediate features compared to ResNet. In contrast, our algorithm reduces the mIoU of all victim models to less than 30%, notably reducing the mIoU of HRNet to 25.35%, which is considerably better than existing methods in generating the transferable samples.

**ADE20k dataset.** The experimental results of the transfer capability on the ADE20k dataset are presented in Table 2. This dataset is more challenging, where the victim model achieves at most 44.08% mIoU (DLV3-R101) even on clean images. Attacks on across-domain and cross-model (HRNet) remain highly challenging on this dataset.

**Table 3: Comparisons of attack mIoU scores (%) on the Cityscapes dataset.**

| Attack | Surrogate model | Black-box Victim model (mIoU ↓) | | | | | |
|---|---|---|---|---|---|---|---|
| | | DLV3-R50 | PSP-R50 | DLV3-R101 | PSP-R101 | UperNet | HRNet |
| Clean Images | - | 79.09 | 77.85 | 77.12 | 78.34 | 79.40 | 78.48 |
| BIA [44] | - | 37.98 | 36.85 | 39.05 | 39.14 | 36.30 | 45.40 |
| CDA [35] | - | 36.27 | 34.52 | 36.97 | 38.28 | 35.53 | 42.92 |
| CDTA [26] | - | 26.72 | 26.85 | 33.07 | 32.69 | 29.76 | 40.74 |
| AGS [41] | - | 26.98 | 26.99 | 35.14 | 31.03 | 33.61 | 38.09 |
| Inkawhich et al. [22] | DLV3-R50 | 21.34 | 22.89 | 26.83 | 25.36 | 22.24 | 35.44 |
| PGD [31] | DLV3-R50 | 24.57 | 24.40 | 35.08 | 31.41 | 25.16 | 41.56 |
| | PSP-R50 | 24.05 | 23.82 | 34.81 | 30.94 | 26.52 | 40.76 |
| FDA [14] | DLV3-R50 | 21.50 | 21.52 | 32.24 | 28.04 | 24.93 | 39.17 |
| | PSP-R50 | 21.56 | 21.55 | 31.39 | 29.38 | 23.94 | 38.68 |
| PFFAA (Ours) | DLV3-R50 | 12.80 | **11.79** | 13.88 | **12.93** | **11.21** | **15.78** |
| | PSP-R50 | **12.79** | 12.09 | **12.49** | 13.85 | 11.67 | 17.14 |

**Table 4: Ablation study of different components on the Pascal VOC2012 dataset. The surrogate model is DLV3-R50.**

| Baseline | Components | | Black-box Victim model (mIoU ↓) | | | | | |
|---|---|---|---|---|---|---|---|---|
| | FAA | PFA | DLV3-R50 | PSP-R50 | DLV3-R101 | PSP-R101 | UperNet | HRNet |
| ✓ | | | 76.17 | 76.78 | 78.70 | 78.47 | 77.10 | 75.87 |
| | ✓ | | 35.21 | 34.51 | 36.22 | 38.47 | 38.91 | 39.22 |
| | | ✓ | 32.99 | 32.29 | 34.66 | 33.77 | 35.24 | 35.64 |
| | ✓ | ✓ | **22.96** | **23.93** | **23.31** | **24.87** | **20.61** | **25.91** |

**Table 5: Ablation study of the order of PFA and FFA on the Pascal VOC2012 dataset. FAA → PFA implies conducting the amplitude attack first, followed by PFA, and PFA → FAA implies conducting PFA first, followed by FAA.**

| Attack | Surrogate model | Black-box Victim model (mIoU ↓) | | | | | |
|---|---|---|---|---|---|---|---|
| | | DLV3-R50 | PSP-R50 | DLV3-R101 | PSP-R101 | UperNet | HRNet |
| Clean Images | - | 76.17 | 76.78 | 78.70 | 78.47 | 77.10 | 75.87 |
| PFA → FAA | DLV3-R50 | 27.16 | 28.33 | 30.21 | 29.64 | 32.15 | 33.49 |
| | PSP-R50 | 26.33 | 27.12 | 29.99 | 28.74 | 31.17 | 32.82 |
| FAA → PFA | DLV3-R50 | 22.96 | 23.93 | **23.31** | 24.87 | 20.61 | 25.91 |
| | PSP-R50 | **22.33** | **21.58** | 24.14 | **22.91** | **20.51** | **25.35** |

For instance, FDA and PGD only reduce mIoU to 24.28% and 27.44%. Notably, our algorithm not only performs far better than other methods in all victim models, but also reduces the mIoU of the victim HRNet to 13.55%, exceeding the benchmark by 10.73%.

**Cityscapes dataset.** Table 3 presents the experimental results on the Cityscapes dataset, which focuses on city scenarios and differs notably from the other datasets in terms of domains. The victim model achieves impressive performance on clean images. Attackers designed for cross-domain attacks also show better performance on this dataset, e.g., CDTA achieves a performance degradation of 52.37% on the victim model DVL3-R50. Given the substantial domain difference, FDA outperforms PGD by approximately 2%-3%. Moreover, the attack of AGS on this dataset is more effective. Compared with them, our method showcases superior attack performance again, particularly on the victim HRNet, surpassing the state-of-the-art result by 19.66%. This demonstrates the advantage of PFFAA to generate transferable adversarial samples in the face of different datasets and different models.

## 4.3 Ablation Study

**Ablation study of each component.** To demonstrate the effectiveness of each component, we conduct corresponding ablation experiments, and the results are shown in Table 4. Initially, the results on clean images are established as the baseline. Subsequently, we introduce PFA on Fourier space, which is independent of the surrogate model. Remarkably, the amplitude attack demonstrated significant efficacy across various black-box victim models. Hence, replacing the amplitude information of test images proves to be highly effective. Then, using only FAA, we observe evident success across different victim models by leveraging class relationships and prototype features. It is worth noting that PFA, as a feature space attack method, has stronger cross-architectural capabilities. Finally, the best results are achieved by combining these two attacks within our approach. This experiment emphasizes the necessity and effectiveness of the proposed components.

**Order of the two attacks.** We investigate the impact of changing the order of the two attack strategies (PFA and FAA) on the

Table 6: Ablation study on the number of queries. The surrogate model is DLV3-R50 in this experiment.

| Number | Black-box Victim model (mIoU ↓) | | | | | |
|---|---|---|---|---|---|---|
| | DLV3-R50 | PSP-R50 | DLV3-R101 | PSP-R101 | UperNet | HRNet |
| Baseline | 76.17 | 76.78 | 78.70 | 78.47 | 77.10 | 75.87 |
| 0 | 35.66 | 36.99 | 38.42 | 39.02 | 37.71 | 47.15 |
| 10 | 35.40 | 36.08 | 37.40 | 38.41 | 35.92 | 44.95 |
| 50 | 33.16 | 34.16 | 36.29 | 35.81 | 33.75 | 39.90 |
| **100** | 32.85 | 32.29 | 34.66 | 33.77 | 32.24 | 35.64 |
| 200 | 32.91 | 32.40 | 34.15 | **33.52** | 32.19 | 35.10 |
| 400 | **32.57** | **32.15** | **33.94** | 33.58 | **31.99** | **35.07** |

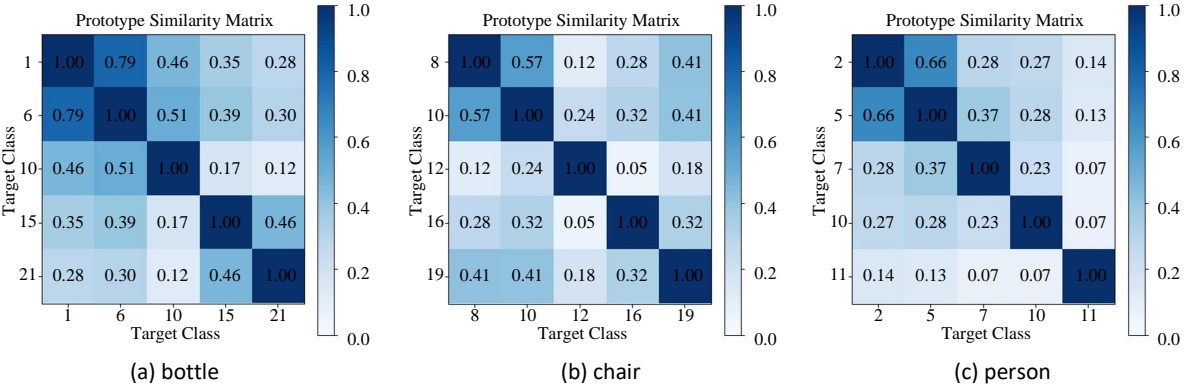

(a) bottle                                   (b) chair                                   (c) person

Figure 3: Visualizing feature prototype similarity on the Pascal VOC2012 dataset. DLV3-R50 is used as the surrogate model, and the victim model is PSP-R50.

effectiveness of the proposed attack. The results, which are listed in Table 5, indicates that applying the frequency alteration attack followed by PFA results in better performance, which aligns with our chosen sequence. This trend primarily arises from the non-adversarial nature of the frequency alteration attack. In other words, while FAA can target unified domain information, it lacks the feedback to fine-tune the image. Conversely, PFA adjusts the images after the frequency alteration attack based on feedback, further enhancing the outcomes.

**Number of queries.** Table 6 presents experimental results of different query numbers on the Pascal VOC2012 dataset. To observe the impact of $n$ more effectively, FAA is not utilized in this experiment, resulting in slightly lower attack effectiveness compared with the complete method. Firstly, when $n = 0$, it means a direct feature distribution attack on the surrogate model. In this case, as there is a lack of relationship between the classes, the mIoU only decreases to 36.99% for the victim model PSP-R50. Subsequently, we gradually increase the number of query images, and relationships between the source classes and the target classes are gradually established. Accordingly, the attack effectiveness gradually improves, although this improvement is decreasing. Even between $n = 400$ and $n = 100$, the effect of the attack decreases for the victim models DLV3-R50 (-0.28%) and PSP-R50 (-0.14%). In addition, as the number of queries gradually increases, there is a potential risk of being detected by the host system. As a trade-off, we set the number of queries to 100. We utilize this setting in our experiments to evaluate the attack effectiveness and the risk of being detected by the host system.

Table 7: Comparisons of attack mIoU scores (%) under different $\epsilon$ on the Pascal VOC2012 dataset. The surrogate model is DLV3-R50 for all methods.

| $\epsilon$ | Attack | Black-box Victim model | |
|---|---|---|---|
| | | DLV3-R50 | PSP-R50 |
| 8 | AGS [41] | 64.97 | 65.27 |
| | Inkawhich et al. [22] | 44.77 | 43.65 |
| | PGD [31] | 57.16 | 58.03 |
| | FDA [21] | 46.66 | 47.10 |
| | PFFAA (Ours) | **37.70** | **38.56** |
| 16 | AGS [41] | 54.28 | 53.50 |
| | Inkawhich et al. [22] | 32.38 | 31.64 |
| | PGD [31] | 38.59 | 38.83 |
| | FDA [21] | 35.66 | 36.99 |
| | PFFAA (Ours) | **22.96** | **23.93** |
| 32 | AGS [41] | 36.67 | 35.57 |
| | Inkawhich et al. [22] | 20.89 | 21.42 |
| | PGD [31] | 28.69 | 28.83 |
| | FDA [21] | 22.73 | 22.58 |
| | PFFAA (Ours) | **8.35** | **8.82** |

**Study of perturbation budgets.** To compare the performance of PFFAA under different perturbation budgets $\epsilon$, Table 7 shows the results of several attackers. Larger perturbation budgets offer greater flexibility to modify image pixels, thereby enhancing the

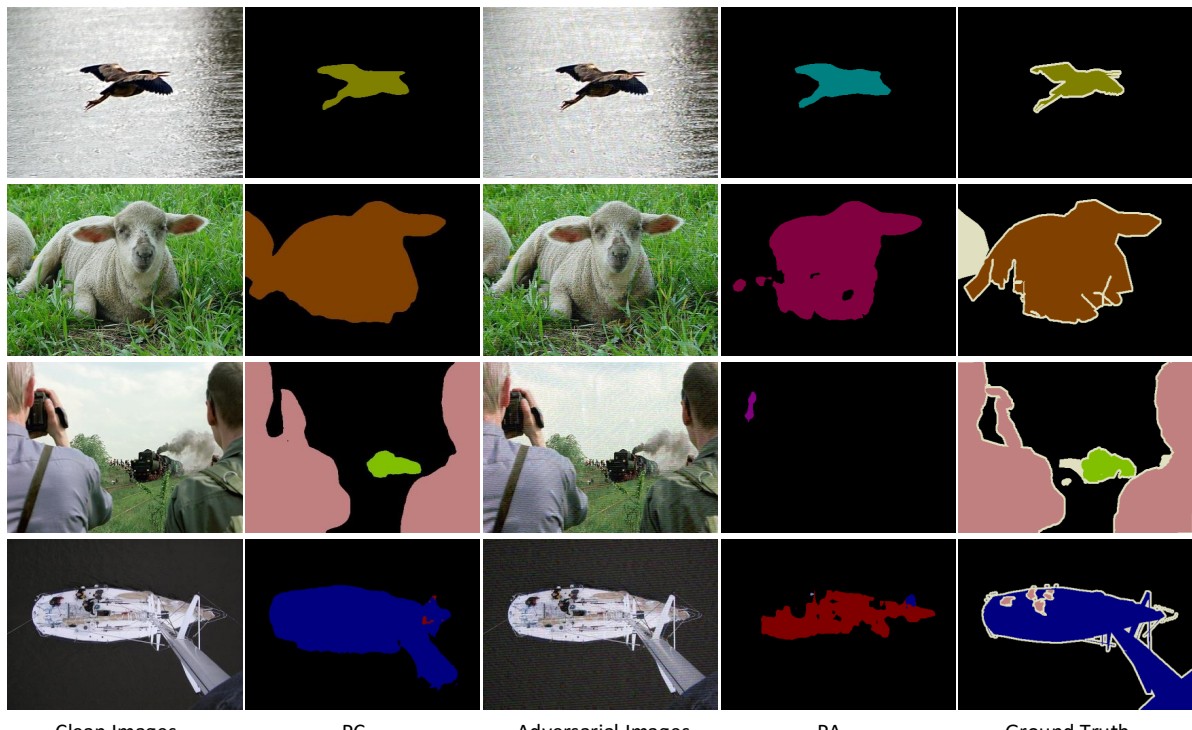

| Clean Images | PC | Adversarial Images | PA | Ground Truth |

**Figure 4: Visualization results of some adversarial examples generated by PFFAA that successfully attack to fool the model on Pascal VOC2012. The victim model is PSP-R50, and the surrogate model is DLV3-R50. 'PA' and 'PC' mean the output of the prediction by the victim model of adversarial images generated by PFFAA and clean images, respectively.**

success rate of the attack. However, such modifications also become more susceptible to detection. The table shows significant enhancement in the attack effectiveness of various methods as the perturbation budget grows. For instance, the FDA reduces mIoU from 46.66% to 22.73% on the victim model DLV3-R50 as the perturbation budget increases from 8 to 32. It is worth noting that PFFAA consistently achieves the best attack results under different perturbation budgets. Specifically, it reduces the mIoU to 8.35% and 8.82% on the victim models (DLV3-R50 and PSP-R50), respectively, at a budget of 32. Experimental results show that PFFAA has significant advantages in cross-domain semantic segmentation attacks.

### 4.4 Qualitative Evaluation

**Feature visualization.** Fig. 3 illustrates the similarity between prototypes with features from multiple source classes, which correspond to the same target class on the Pascal VOC2012 dataset. We show three target classes: (a) *"bottle"*, (b) *"chair"* and (c) *"person"*, respectively. For semantic segmentation, a single target class may correspond to multiple source classes. It is noticeable that the intermediate features of multiple source classes belonging to a single target class are not entirely similar. In fact, they may exhibit substantial differences (less than 0.2 similarity). Therefore, if only FDA is used for feature perturbation, although it can move the features of the adversarial image away from the clean image features, it might bring them closer to another source class corresponding to

the same target class, thus failing to achieve the intended attack effect. These results also validate the need for designing PFA.

**Attack result visualization.** Fig. 4 showcases that the cross-domain adversarial images generated by the surrogate model (DLV3-R50) can effectively deceive the black-box victim model PSP-R50 in the Pascal VOC2012 dataset. The trained model predicts the foreground class as a wrong class on the adversarial image. This misclassification is due to the perturbations introduced, which effectively manipulate the decision boundary of the model to interpret foreground features as belonging to other classes.

## 5 CONCLUSIONS

In this paper, we introduced a novel cross-domain attack strategy explicitly designed for semantic segmentation, called Prototype-based Feature and Frequency Alteration Attack (PFFAA). PFFAA empowers a surrogate model to effectively deceive a black-box victim model without the target data. It consists of two parts: the Prototype-based Feature Attack (PFA) and the Frequency Alteration Attack (FAA). The former resolves feature space perturbations that might redirect features to other features related to the same target class, and the latter involves substituting frequency information from images used to train the surrogate model into the frequency domain of the test images, altering texture and structure. Experimental results across diverse datasets and victim models affirm that PFFAA achieves top-performing cross-domain attacks without the need for target data.

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
