# OpenReview forum: "PFFAA: Prototype-based Feature and Frequency Alteration Attack for Semantic Segmentation"
_acmmm.org/ACMMM/2024/Conference — MM2024 Poster_

### Official Review · Reviewer_GkEq · 2024-05-11

**Rating:** 3
**Confidence:** 3

**Summary:**

This paper focus on the cross-domain black-box adversarial attack on semantic segmentation models. Specifically, the authors propose to use feature prototypes for each class to perform the attack. To further narrow the gap between the source-domain and target-domain samples, a frequency feature manipulation on images is leveraged. Experiments on three common datasets states the effectiveness of the method.

**Strengths:**

- Introducing frequency analysis into black-box adversarial attacks seems to be a novel idea.
- The method has achieved SOTA performance in comparison with the selected baselines.

**Limitations:**

- The threat model is unclear. It will be more friendly to readers if the authors could specify which knowledge an attacker has, e.g., the number of classes of the target domain, the prediction probability of the target model, etc. This determines how hard the "black-box attack" is.
- The query budget comparison with other baselines is not shown. From Table 6, we can learn about the ablation of the proposed method on number of queries. It is suggested to show whether the proposed method can further reduce the query number requirement.
- The presentation of the paper can be improved. For instance, in Fig.1 the difference between FDA and PFA is not clear enough.

### Some other questions.

- Since the content of COCO 2017 has different similarities with those in Pascal VOC 2012, ADE20K, and Cityscapes, I wonder if there will be difference in performance when attacking models for the latter three datasets with COCO as the source domain dataset.

**Suitability:**

2

---

### Official Review · Reviewer_yVpq · 2024-05-13

**Rating:** 4
**Confidence:** 2

**Summary:**

The paper introduces PFFAA, a cross-domain adversarial attack method for semantic segmentation that allows a surrogate model to deceive a black-box victim model without target data access. PFFAA includes Prototype-based Feature Attack (PFA) to address feature space perturbations and Frequency Alteration Attack (FAA) to modify image texture and structure. Experiments demonstrate PFFAA's effectiveness in generating transferable adversarial samples, outperforming state-of-the-art methods.

**Strengths:**

- This paper targets the cross-domain attack, which is a promising field.
- Mix the methods of prototype, the Fourier amplitude and the correlation matrix. The PFA part is interesting.
- The proposed method achieves state-of-the-art performance in many scenarios with a moderate number of queries.

**Limitations:**

- This paper does not compare its proposed method with attacks proposed against semantic segmentation. For example, EBAD and Rony et al. should be evaluated and compared.
- Line 402, how to get the source amplitude?
- The ablation study of each component is not very clear.
- In Figure 4, the visualization result of some adversarial examples is not good. The perturbation seems easy to indentify for humans.

**Suitability:**

2

---

### Official Review · Reviewer_3dbm · 2024-05-24

**Rating:** 4
**Confidence:** 3

**Summary:**

This paper introduces a novel cross-domain attack method, Prototype-based Feature and Frequency Alteration Attack (PFFAA), which is specifically tailored for semantic segmentation. This method represents an advancement in adversarial attacks by allowing the surrogate model to deceive a black-box victim model without requiring access to the target domain.

**Strengths:**

（1）	The paper conducts extensive experiments on multiple datasets (Pascal VOC2012, ADE20k, Cityscapes) and various victim models (PSPNet, DeepLabV3, UperNet, HRNet), providing robust validation of the PFFAA method's effectiveness. Experimental results demonstrate its superior performance and efficiency.
（2）	The methodology is well-structured, with clear descriptions of the PFA and FAA components, supported by diagrams and pseudo-code to aid understanding.
（3）	The paper provides theoretical insights into the effectiveness of the PFFAA method, demonstrating how the approach guarantees successful attacks through precise control of feature and frequency alterations.

**Limitations:**

（1）The advantages of the experimental results over previous approaches are not significantly pronounced and require further in-depth analysis. A more detailed comparison and discussion of the performance metrics would help highlight the benefits of the proposed method and provide a clearer understanding of its strengths relative to existing methods.
（2）There is a lack of defense suggestions for the Prototype-based Feature and Frequency Alteration Attack (PFFAA), with no specific measures provided to counter or mitigate the impact of this attack. This raises concerns about the security of practical applications.

**Suitability:**

3

---

### Meta-Review · Area_Chair_qXxL · 2024-06-28

**Recommendation:** Accept (Poster)
**Confidence:** 4

**Metareview:**

This paper introduces a novel cross-domain adversarial attack method against semantic segmentation. It exploits a frequency feature manipulation on images to narrow the gap between the source-domain and target-domain samples. Reviewers generally consider the method novel and effective, and the paper is well-written to a large extent. Although some issues were raised, critical concerns were well addressed during the rebuttal.

We hope the authors can conduct a more in-depth analysis of the effectiveness of the proposed method, e.g., how the prototypes help the attacks and why such prototypes can transfer between different data domains. We urge the authors to adjust the paper as claimed in the rebuttal.